# Optimization for a Multi-Constraint Truck Appointment System Considering Morning and Evening Peak Congestion

Bowei Xu [1],*, Xiaoyan Liu [1], Yongsheng Yang [1], Junjun Li [2] and Octavian Postolache [3]

1 Institute of Logistics Science and Engineering, Shanghai Maritime University, Shanghai 201306, China; 201830510018@stu.shmtu.edu.cn (X.L.); yangys@shmtu.edu.cn (Y.Y.)
2 College of Merchant Marine, Shanghai Maritime University, Shanghai 201306, China; lijj@shmtu.edu.cn
3 ISCTE, Lisbon University Institute, 1649-026 Lisboa, Portugal; opostolache@lx.it.pt
* Correspondence: bwxu@shmtu.edu.cn

**Abstract:** Gate and yard congestion is a typical type of container port congestion, which prevents trucks from traveling freely and has become the bottleneck that constrains the port productivity. In addition, urban traffic increases the uncertainty of the truck arrival time and additional congestion costs. More and more container terminals are adopting a truck appointment system (TAS), which tries to manage the truck arrivals evenly all day long. Extending the existing research, this work considers morning and evening peak congestion and proposes a novel approach for multi-constraint TAS intended to serve both truck companies and container terminals. A Mixed Integer Nonlinear Programming (MINLP) based multi-constraint TAS model is formulated, which explicitly considers the appointment change cost, queuing cost, and morning and evening peak congestion cost. The aim of the proposed multi-constraint TAS model is to minimize the overall operation cost. The Lingo commercial software is used to solve the exact solutions for small and medium scale problems, and a hybrid genetic algorithm and simulated annealing (HGA-SA) is proposed to obtain the solutions for large-scale problems. Experimental results indicate that the proposed TAS can not only better serve truck companies and container terminals but also more effectively reduce their overall operation cost compared with the traditional TASs.

**Keywords:** multi-constraint truck appointment system; morning and evening peak congestion; hybrid genetic algorithm and simulated annealing; gate and yard congestion; mixed integer nonlinear programming

## 1. Introduction

Over 90% of international trade is transported by ship. Container shipping is one of the dominant transportation modes. In 2018, the top 20 container ports in the word completed a container throughput of 340 million TEU (Twenty feet Equivalent Units) and reached an increase of 3.8% over the last year. Thus, container ports are a distribution center for container cargoes and an important support for international trade. In the era of globalization, high-quality container port logistics can promote countries' stability and development.

However, congestion is ubiquitous in container ports. The impact of port congestion is multifaceted. (a) Port congestion directly affects the economic benefits of participants in the port supply chain (Namboothiri and Erera, 2008) [1], resulting in longer ship docking time, higher transportation and operation costs, more difficult port production management, and larger backlogs of goods. (b) If there are not proper governance measures implemented, congestion may spread to surrounding ports as time goes by. For example, the congestion in Shanghai Port in April 2017 spread to Qingdao Port and Ningbo Zhoushan Port [2]. (c) Increased emissions [3]) of congested ports affect city image and residents' quality of life [4,5]. Congestion has become the bottleneck that constrains port productivity and

restricts ports' sustainable development. Therefore, it is crucial and urgent to solve port congestion problems.

To alleviate port congestion, many studies have proposed solutions. Regan and Golob [6] found that the efficiency of maritime transport depended on the smooth operation of inland transport. Truck companies can use information technology to reduce delays inside and outside the port. Knowing the time when a truck arrives at the port can improve the efficiency of port operations. Ozbay et al. [7] found that the Port Authority of New York and New Jersey charging plan successfully made the arrival time of trucks more uniform. Dekker et al. [8] introduced the concept of the Chassis Exchange Terminal at the Container Terminal in Rotterdam. The main idea was to provide an off-dock terminal. The truck would leave the outbound container on the trailer and take the trailer with the inbound container when the harbor terminal was not congested. Van et al. [9] evaluated the impact of the arrival time of the truck on reducing the inefficient movement of the yard crane. Grubisic et al. [10] studied ports located near the city center to determine the key traffic parameters, the queue length on the road to the container terminal, and the parking delay on the main city corridor, which had negative organizational and environmental impacts on current and future traffic demand. However, the overall costs of the truck companies and container terminals were not considered. The port needs to process many orders every day, which is also a cause of congestion. Pandian [11] proposed a procedure for managing large-scale orders to develop a flexible plan to meet customer requirements.

To reduce gate and yard congestion, more and more container terminals (such as Los Angeles Port and Long Beach Port in the United States, Vancouver Port in Canada, and Tianjin Port in China) use a Truck Appointment System (TAS) as it is called in the U.S., or vehicle booking system (VBS) in other parts of the world [12]. Giuliano and O'Brian [13] mentioned, for the first time, the implementation of a TAS in the ports of Los Angeles and Long Beach. The main idea of TAS is that the truck company pre-arranges the working hours of the truck, giving each truck a designated time [14]); the port will pre-determine the allocation of yard equipment and goods [15,16]. TAS is one of the best and most common communication methods between truck companies and container terminals [17,18]. Ramírez-Nafarrate et al. [19] proposed a discrete event simulation model and a heuristic process to analyze the potential configuration of TAS and evaluate its impact on yard operations, especially in terms of reducing container heavy handling and truck turn time. However, this research is limited to the impact of TAS on the port, especially the internal port, and does not consider the cost impact of the program on trucks. To this end, many studies have attempted to determine the best design for TAS. Phan et al. [20] considered the impact of the truck on changing the arrival time and established a multi-constraint TAS model combining mathematical formulas and decentralized decision-making to support the negotiation between the truck company and the terminal company, making the time of the arrival of the trucks more uniform. However, it did not consider the impact of urban traffic on the arrival time of trucks nor did it consider the minimum costs of truck companies and port companies. Chen et al. [21] proposed a method called "ship-related time-window" to control the arrival of the trucks. However, it only considered the congestion of the truck at the gate and the corresponding cost and did not consider the overall cost of the truck companies and the container terminals. Mohammad et al. [22] established a mixed integer nonlinear model, which can serve both the truck company and the terminal company and can alleviate the congestion of the truck at the gate on the basis of effectively reducing transportation costs. However, the environment is idealized, and the impact of urban traffic on the time of truck arrival at the port is not considered. Urban traffic increases the uncertainty of the truck arrival time and additional congestion costs. However, it seems that there are few studies to include urban traffic in the scope of TAS. Kot [23] discussed the determination of road transport costs and related issues based on the transport services of road transport in Poland and certain EU countries and quantified the specific costs of road transport.

In summary, previous studies on TAS mainly considered the queue length of trucks at the gate, and few studies considered the interests of both truck companies and port companies, and the impact of morning and evening peak traffic on the time of truck arrival at the port. In an attempt to fill these research gaps, this study serves as a starting point in explicitly considering both the impact of urban traffic on TAS and the overall operation costs of the truck companies and the container terminals. Therefore, the aim of the study is to develop a higher-quality TAS with improved rationality and effectiveness. Specifically, such a TAS can better determine the truck's appointment time-window, lessen the impact of adjustment on the truck company's expected appointment plans, mitigate the queue time of the truck at gates, and meet the order demand of the container terminals. In order to meet these aims, we proposed a multi-constraint TAS model based on mixed integer nonlinear programming (MINLP) to determine the best appointment plan for each truck. Lingo software is used to solve small and medium-sized problems, a hybrid genetic algorithm and simulated annealing method is proposed to solve large-scale problems. Experimental results successfully demonstrate that the multi-constraint TAS proposed in this work outperforms traditional TASs. The proposed model and method not only minimize its overall operation cost but also improve the operation efficiency.

The rest of the study is organized as follows: the problem definition and a multi-constraint TAS model are provided in Section 2. A hybrid genetic algorithm and simulated annealing method is presented in Section 3. The computational results and discussions are shown in Section 4. Finally, Section 5 concludes the work.

## 2. Problem Definition and a Multi-Constraint TAS Model

The problem may be formally defined as follows. The truck company submits the appointment request for the next day before 5 p.m. every day; at the same time, the terminal company submits the quota for each time-window for the next day before 5 p.m. every day. The truck ID and the corresponding container number are required for each truck appointments. The TAS will determine the respective expected time-gap between the consecutive appointments of the truck. The time-gap is assumed to be constant. Once all the input data (appointment requests and quotas) are completed, the TAS finalizes the appointment time-window for each truck, with the aim of minimizing the combined operation costs of the truck and terminal companies. Finally, the TAS sends the best appointment time-window for each truck to the truck company. If the last specified appointment time-window is different from the appointment time-window submitted by the truck company, the truck company will reschedule the truck collection.

The decision process of the truck appointment scheduling from the perspective of the port supply chain is shown in Figure 1. The truck companies and the container companies submit their own appointments and order details to the TAS. The TAS aims to minimize the overall operation cost for both truck companies and port companies. The TAS optimizes the plan, which is finally sent to the truck companies and the container companies respectively. The ultimate objective of the port supply chain is to meet the freight transportation demands of customers.

The main factors affecting the truck appointment are the terminal's acceptable share of appointments, the duration of each time-window, and the time rule of the truck arriving at the port. Considering the impact of the peak period and the arrival volume of the next day, the terminal's acceptable share of appointments is unevenly distributed. The duration of each time-window will affect the arrival distribution of the truck. If it is short, it could be beneficial to terminal operators because they can have a higher control of the truck arrivals in every hour. However, it will reduce the probability that the truck will arrive on time within the scheduled time. Therefore, extending the existing research, this work divides the period from 8:00 a.m. to 6:00 p.m. every day into 10 time-windows. The terminal time-window duration is 1 h. The arrival rule of the truck has a law of changing with time, so the queuing model of non-stationary arrival is used to describe it.

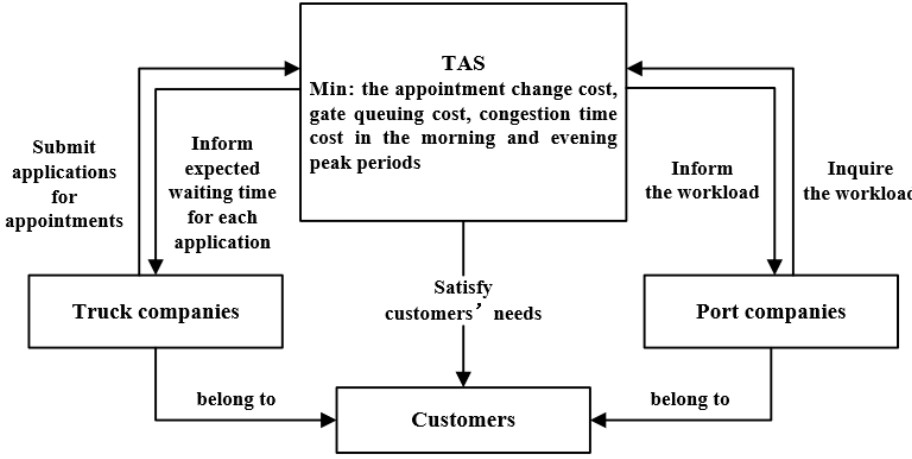

**Figure 1.** Decision process of truck appointment scheduling.

## 2.1. Indices, Parameters and Sets for a Multi-Constrained TAS

### 2.1.1. Indices

| | |
|---|---|
| $w_t$ | Every time-window of the port |
| $i_t$ | Every time-interval of the time-window |
| $h$ | The ID of each truck |
| $f$ | The number of appointments for a truck |
| $t_n$ | A number corresponding to the number of appointments for the next day for each truck |
| $z$ | Truck company's unique number |
| $g$ | Urban road grades |

### 2.1.2. Sets

| | |
|---|---|
| $W_t$ | All terminal time-windows |
| $I_t$ | All time-intervals contained in each terminal time-window |
| $F$ | The number of appointments for the next day for each truck (the number of appointments ranged from 1 to 5) |
| $G_f$ | A truck with all the appointments for the next day |
| $Z$ | All the truck companies |
| $H_z$ | All trucks belonging to truck company z |
| $G$ | All urban road congestion grades |

### 2.1.3. Parameters

| | |
|---|---|
| $N_{w_t}$ | The maximum quota for each terminal time-window |
| $D_{hft_n}$ | The truck company submits the truck arrival time-window |
| $\varphi w_t$ | The maximum service efficiency for each terminal time-window |
| $a$ | The number of time-intervals in a terminal time-window |
| $\xi$ | Coefficient of variance of gate service time |
| $W_l$ | The penalty value when the difference between the actual arrival time-window of the truck and the adjacent time-window is larger than the expected |
| $W_s$ | The penalty value when the difference between the actual arrival time-window of the truck and the adjacent time-window is smaller than the expected |

| | |
|---|---|
| $W_d$ | The penalty weight for early arrival trucks |
| $W_n$ | The penalty weight for late arrival trucks |
| $W_q$ | The penalty value of the queue length of a truck queuing at a gate |
| $N_{dz}$ | The number of truck appointment by truck company $z$ |
| $S_z$ | The initial cost threshold of truck company $z$ |
| $dt_{dhft_n}$ | Time-gap between two adjacent appointments of truck $h$ with $f$ appointments for the expected appointment schemes provided by the truck company |
| $\sigma$ | Loss coefficient of economic cost when suffering traffic jam |
| $C_g$ | Length of $g$ grade road |
| $H_g$ | The number of trucks on roads with different levels of congestion |
| $V_g$ | Critical speeds of trucks on roads with different levels of congestion |
| $V'_g$ | Trucks traveling at normal speed on different classes of roads |

### 2.1.4. Decision Variables

| | |
|---|---|
| $R_{i_t}$ | The average truck arrival rate at $i_t$ |
| $L_{i_t}$ | The average queue length of trucks at the gate at $i_t$ |
| $V_{i_t}$ | The average departure rate from terminal gate to the terminal yard at $i_t$ |
| $dt_{hft_n}$ | Time-gap between two adjacent appointments of truck $h$ with $f$ appointments for the actual appointment schemes confirmed by port company |
| $Z_{hft_n}$ | $dt_{dhft_n} - dt_{hft_n}$ |
| $Z'_{hft_n}$ | $dt_{hft_n} - dt_{dhft_n}$ |
| $Y_{hft_n}$ | The difference between the actual and expected time-windows of the truck arrivals |
| $Y'_{hft_n}$ | The difference between the expected and actual time-windows of the truck arrivals |
| $C_z$ | The appointment change cost of truck company $z$ |

### 2.2. A Multi-Constraint TAS Model

Considering the discreteness of terminal time-window, relevant decision variables are defined in the multi-constraint TAS model. The decision variable explains the arrival of a specific truck at a certain time-window at the terminal (Phan and Kim, 2016).

$$M_{hft_nw_t} = \left\{ \begin{array}{l} 1 \quad \text{If truck } h \text{ with } f \text{ appointment (s) has its } t_n{}^{th} \text{appointment at time} - \text{window } w_t \\ 0 \qquad\qquad\qquad\qquad\qquad\qquad Otherwise \end{array} \right. \tag{1}$$

$$\min \sum_{z \in Z} C_z + W_q \sum_{i_t \in I_t} \left( L_{i_t} + \frac{R_{i_t} - V_{i_t}}{2} \right) + \sigma \sum_{L \in \{1,2.3\}} C_L \cdot \left( \frac{1}{V_L} - \frac{1}{V'_L} \right) \cdot M_L \tag{2}$$

Subject to:

$$\begin{aligned} C_z = \;& W_l \sum_{h \in H_z} \sum_{f \in F} \sum_{t_n \in G_f} Z_{hft_n} + W_s \sum_{h \in H_z} \sum_{f \in F} \sum_{t_n \in G_f} Z'_{hft_n} \\ & + W_d \sum_{h \in H_z} \sum_{f \in F} \sum_{t_n \in G_f} Y_{hft_n} + W_n \sum_{h \in H_z} \sum_{f \in F} \sum_{t_n \in G_f} Y'_{hft_n} \quad \forall z \in Z \end{aligned} \tag{3}$$

$$Z_{hft_n} \geq 0 \; \forall z \in Z, \; \forall h \in H_z, \; \forall f \in F \backslash \{1\}, \; \forall t_n \in G_f \backslash \{f\} \tag{4}$$

$$Z_{hft_n} \geq dt_{hft_n} - dt_{dhft_n} \; \forall z \in Z, \; \forall h \in H_z, \; \forall f \in F \backslash \{1\}, \; \forall t_n \in G_f \backslash \{f\} \tag{5}$$

$$Z'_{hft_n} \geq 0 \; \forall z \in Z, \; \forall h \in H_z, \; \forall f \in F \backslash \{1\}, \; \forall t_n \in G_f \backslash \{f\} \tag{6}$$

$$Z'_{hft_n} \geq dt_{dhft_n} - dt_{hft_n} \; \forall z \in Z, \; \forall h \in H_z, \; \forall f \in F \backslash \{1\}, \; \forall t_n \in G_f \backslash \{f\} \tag{7}$$

$$Y_{hft_n} \geq 0 \; \forall z \in Z, \; \forall h \in H_z, \; \forall f \in F \backslash \{1\}, \; \forall t_n \in G_f \backslash \{f\} \tag{8}$$

$$Y_{hft_n} \geq \sum_{w_t \in W_t} (w_t \cdot M_{hft_nw_t}) - D_{hft_n} \; \forall z \in Z, \; \forall h \in H_z, \; \forall f \in F \backslash \{1\}, \; \forall t_n \in G_f \backslash \{f\} \tag{9}$$

$$Y'_{hft_n} \geq 0 \ \forall z \in Z, \ \forall h \in H_z, \ \forall f \in F \backslash \{1\}, \ \forall t_n \in G_f \backslash \{f\} \tag{10}$$

$$Y'_{hft_n} \geq D_{hft_n} - \sum_{w_t \in W_t} (w_t \cdot M_{hft_n w_t}) \ \forall z \in Z, \ \forall h \in H_z, \ \forall f \in F \backslash \{1\}, \ \forall t_n \in G_f \backslash \{f\} \tag{11}$$

$$C_z / N_{dz} \leq S_z \ \forall z \in Z \tag{12}$$

$$S_z = \varepsilon + \gamma \cdot b^{\frac{1}{N_{dz}}} \ \forall z \in Z \tag{13}$$

$$dt_{hft_n} = \sum_{w_t \in W_t} (w_t \cdot M_{hf(t_n+1)w_t}) - \sum_{w_t \in W_t} (w_t \cdot M_{hft_n w_t}) \ \forall z \in Z, \ \forall h \in H_z, \ \forall f \in F \backslash \{1\}, \ \forall t_n \in G_f \backslash \{f\} \tag{14}$$

$$dt_{dhft_n} = D_{hf(t_n+1)} - D_{hft_n} \ \forall z \in Z, \ \forall h \in H_z, \ \forall f \in F \backslash \{1\}, \ \forall t_n \in G_f \backslash \{f\} \tag{15}$$

$$\sum_{w_t \in W_t} M_{hft_n w_t} = 1 \ \forall z \in Z, \ \forall h \in H_z, \ \forall f \in F \backslash \{1\}, \ \forall t_n \in G_f \backslash \{f\} \tag{16}$$

$$\sum_{h \in H_z} \sum_{f \in F} \sum_{t_n \in G_f} M_{hft_n w_t} \leq N_{w_t} \ \forall w_t \in W_t \tag{17}$$

$$w_t \cdot \sum_{w_t \in W_t} M_{hft_n w_t} \leq w_t \cdot \sum_{w_t \in W_t} M_{hf(t_n+1)w_t} \ \forall w_t \in W_t \tag{18}$$

$$R_{i_t} = \sum_{h \in H_z} \sum_{f \in F} \sum_{t_n \in G_f} M_{hft_n w_t} \ \forall i_t \in I_t, \ \forall w_t \in W_t \tag{19}$$

$$V_{i_t} \leq \varphi_{w_t} \cdot \frac{L_{i_t} + 1 - \sqrt{L_{t_i}^2 + 2\xi^2 \cdot L_{i_t} + 1}}{1 - \xi^2} \ \forall i_t \in I_t, \ \forall w_t \in W_t \tag{20}$$

$$V_{i_t} \leq L_{i_t} + R_{i_t} \ \forall i_t \in I_t, \ \forall w_t \in W_t \tag{21}$$

$$L_{i_t+1} = L_{i_t} + R_{i_t} - V_{i_t} \ \forall i_t \in I_t, \ \forall w_t \in W_t \tag{22}$$

$$M_{hft_n w_t} = \{0 \ or \ 1\} \ \forall z \in Z, \ \forall h \in H_z, \ \forall f \in F \backslash \{1\}, \ \forall t_n \in G_f \backslash \{f\}, \ w_t \in W_t \tag{23}$$

$$Q_l = \sum_{l \in L} H_l \cdot V_l \ \forall L \in \{1, 2, 3\} \tag{24}$$

Equation (2), which minimizes the overall operation cost of truck companies and port companies, is the objective function of this work. The first term of the objective function represents the total cost for the truck companies to change the appointment schemes. Equations (3) represents the cost consumed by each truck company to change the appointment scheme. The first item of Equation (3) represents the cost loss when the difference between the modified time-window and the adjacent time-window becomes larger after the scheduled time-window of the truck is changed ($X_1$). The second item represents the cost loss when the difference between the modified time-window and the adjacent time-window becomes smaller after the scheduled time-window of the truck is changed ($X_2$). The third item represents the cost of delaying the actual scheduled time-window of the truck ($X_3$). The third represents the cost of the actual appointment time-window of the truck being earlier than expected ($X_4$). Equations (4)–(7) are the linearized form of the maximum difference between the actual time interval and the expected time interval. Equations (8)–(11) are the linearized form of the maximum difference between the actual retention time-window and the expected retention time-window. Equation (12) represents that the corresponding cost increase of each truck company shall not exceed its corresponding threshold. Equation (13) represents the threshold for each truck company (Mohammad et al., 2018), where parameter $\varepsilon$ represents the lowest threshold that the terminal operator wants to apply for all truck companies. Parameter $\gamma$ represents the starting threshold for those truck companies that have relatively few appointments. Parameter $b$ represents the rate of decreasing threshold. This means that the truck company that applies for modification of appointment scheme more often has a smaller threshold, which can reasonably control the modification times

and cost of the truck company. Equation (14) represents the number of time-window differences between two adjacent appointment time-windows in the actual appointment scheme of a truck. Equation (15) represents the number of time-window differences between two adjacent appointment time-windows in the expected appointment scheme of a truck. Equation (16) represents the premise that the appointment scheme satisfies. Equation (17) represents the number of appointments for each time-window should not be greater than the number of orders for changing the time-window. The pre-defined quota is calculated as Equations (25) and (26), where $AQP$ is the average quota per time-window. Except for the final sensitivity analysis, the default time-window is 10 per day. Equation (26) is the quota setting for each time-window. Equation (18) represents that the order of original appointments for the same truck cannot be changed, that is, the first appointment is still ranked first after the change.

$$AQP = 2\sum_{z \in Z} N_{dz}/10 \tag{25}$$

$$w_t = [0.9AQP, 0.9AQP, 1.1AQP, 1.1AQP, 1.1AQP, 1.1AQP, 1.1AQP, 1.1AQP, 0.9AQP, 0.9AQP] \tag{26}$$

The second term of the objective function represents the waiting cost of the truck at the gate ($X_5$). Equation (19) represents dividing the number of all the trucks arriving in a time-window by the number of time-intervals in the time-window, obtaining the average number of arrivals of the truck in each time-interval. The term time-interval is used only for queue length estimation. The use of time-interval is needed because the Pointwise Stationary Fluid Flow approximation (PSFFA) method used in Equations (19)–(22) requires a shorter duration than time-windows [20,24]. Figure 2 shows the application principle of the PSFFA method, such as that the average queue length of the interval is $L_{i_4} + \frac{R_{i_4} - V_{i_4}}{2}$. The gate truck queue is regarded as M/G/1 queue and is represented by the corresponding queuing function, as in Equation (20). Equations (18) and (19) are used to set the deviation rate from the gate to the container yard at each time interval to the minimum sum of $\varphi_{w_t} \cdot \frac{L_{i_t} + 1 - \sqrt{L_{i_t}^2 + 2\xi^2 \cdot L_{i_t} + 1}}{1 - \xi^2}$ and $L_{i_t} + R_{i_t}$. Equation (22) is used to calculate the queue length for each time-interval. Equation (23) is a representation of the value of the decision variable.

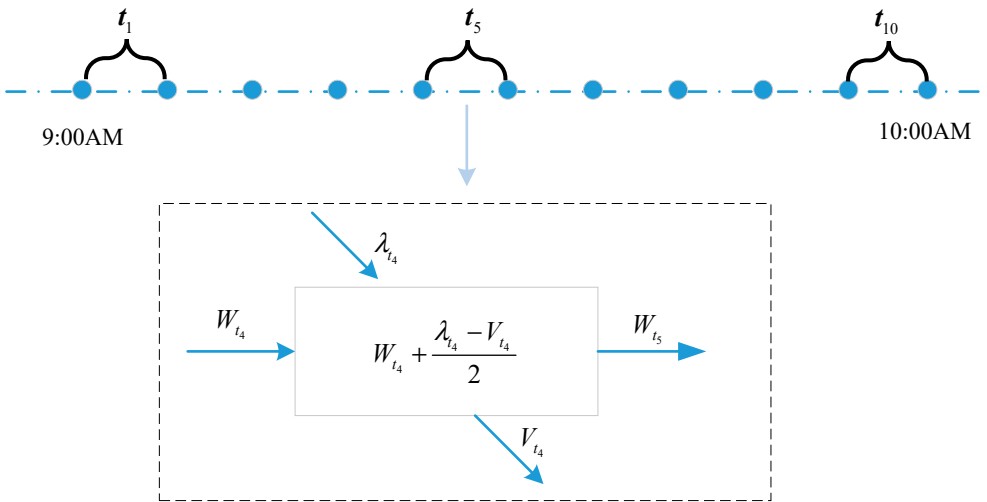

**Figure 2.** The description of the Pointwise Stationary Fluid Flow approximation (PSFFA) method.

The third term of the objective function is the time cost of the morning and evening rush hours encountered by trucks on urban traffic roads ($X_6$), where $\sigma$ is about 2. Urban roads are divided into three levels: highways, trunk roads and secondary trunk roads. According to the actual traffic highway network, the time costs of trucks with different road levels under different congestion intensity are calculated respectively. The critical speeds of congested trucks in different time states refer to the classification standard of

traffic status of different types of roads in the city. Equation (24) calculates the number of trucks at different levels of road congestion.

## 3. A Hybrid Genetic Algorithm and Simulated Annealing Method

The truck appointment scheduling problem is an NP-hard (NP means non-deterministic polynomial) problem, which belongs to the extension of the multiple Traveling Salesman Problem with Time-Windows (m-TSPTW) [25,26]. The increased waiting time of the peak time increase the difficulty of solving the problem, making the multi-constrained scheduling problem of the truck appointment system more complicated.

Lingo software is used to solve mixed integer nonlinear programming problem, and solutions to the small and medium scale problems are obtained. The lingo software is prone to being "out of memory" in solving large-scale problems.

Genetic algorithms [27] are well close to the best in a short span of generations, but many functions may be required to achieve convergence. The simulated annealing algorithm [28] requires a long calculation time to get close to the best, but for local searches, the algorithm is faster and more efficient. To make full use of the global search ability of the genetic algorithm and the local search ability of the simulated annealing algorithm, this work proposes a hybrid genetic algorithm and simulated annealing (HGA-SA), as shown in Figure 3a. A new iterative method is proposed, and the mutation probability is autonomously adjusted according to the evolutionary algebra, which reduces the solution time of the algorithm. The steps of the HGA-SA are as follows:

Step 1: Encoding and initialization. The population individuals are randomly initialized under constraints (16), (17), and (25), and the population number is N (its value is 100). The initial parameter of the threshold Equation (13) is set to $\varepsilon = 8$, $\gamma = 4\varepsilon$, $b = 1.35$. The representation of each individual is shown in Figure 3b. For each individual, there are 10 random numbers (one working day is divided into 10 time-windows), and the sum of 10 random numbers is certain, both are 2AQP. Each random number represents the maximum quota for a time-window, and each individual represents a solution for an appointment allocation.

Step 2: Calculation of the fitness value. The cost of each individual according to the objective function is calculated, and 1000 times its reciprocal is taken as the fitness value of the individual:

$$A = 1000 / \sum_{z \in Z} C_z + W_q \sum_{i_t \in I_t} \left( L_{i_t} + \frac{R_{i_t} - V_{i_t}}{2} \right) + \sigma \sum_{L \in \{1,2,3\}} C_L \cdot \left( \frac{1}{V_L} - \frac{1}{V'_L} \right) \cdot M_L \qquad (27)$$

Step 3: Genetic selection. The best individuals of the previous generation are put directly into the mating pool, and other children are selected from the parent by roulette.

Step 4: Genetic crossover. A sub-population of the population is produced by the intersection of genes. Since the multi-constraint TAS model needs to ensure that the genetics of each individual and the constant (the total quota is unchanged), this work uses a two-point crossover operator, as shown in Figure 3c. First, a random number is generated in the range [0, 1]. If the random number is less than the predefined crossover rate (its value is 0.6), the crossover operator is run. Two intersections, *a* and *b*, are randomly generated. The first *p* gene of the sub-individual 1 are composed of the first *p* gene of the parent individual 1, the genes at the *q-p* positions are composed of the *q-p* positions' genes of the parent individual 2, and the 10-*q* genes are composed of the 10-*q* genes of the parent individual 1. Sub-individual 2 consists of the remaining genes.

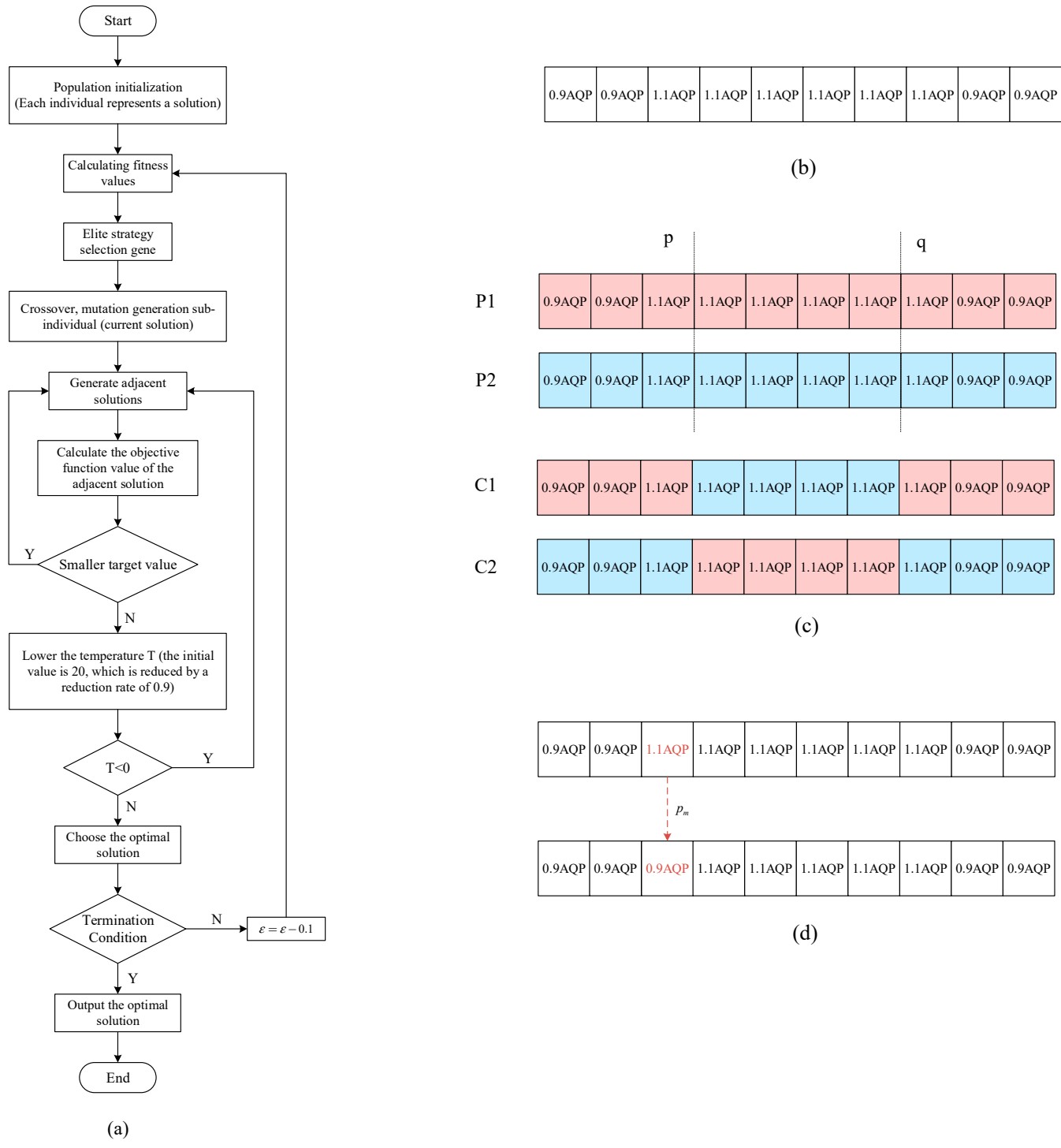

**Figure 3.** Algorithm correlation diagram ((**a**): The HGA-SA flow chart; (**b**): Chromosome representation; (**c**): Genetic crossover; (**d**): Genetic mutation).

Step 5: Genetic mutation. This work uses non-uniform mutation, as shown in Figure 3d. The appointment quota in each time-window is mutated. The change amount is set to a random integer between [−0.2AQP, 0.2AQP] and opposite to each other. The generated new individual satisfies the constraint condition in the multi-constraint TAS model. The probability of mutation $P_m$ is as in Equation (28), where $P_{\max}$ is the maximum

mutation probability, $P_{\min}$ is the minimum mutation probability, and *iter* is the number of current iterations.

$$p_m = p_{\max} - \frac{(p_{\max} - p_{\min})iter}{2000} \tag{28}$$

Step 6: Partial optimization of the solution using a simulated annealing algorithm. In each iteration, an adjacent solution is generated from the current solution as the initial value of the SA (simulated annealing). The target value function value is calculated separately. If the adjacent solution value is smaller, the current solution is replaced by the adjacent solution; otherwise, the adjacent solution is accepted with a certain probability.

Step 7: Determine if the termination condition is met. If the multi-constraint TAS model has no feasible solution, output the current optimal solution; otherwise, decrease the value of parameter $\varepsilon$ ($\varepsilon = \varepsilon - 0.1$) and return to step 3.

## 4. Computational Results and Discussions

In order to demonstrate the feasibility of the multi-constraint TAS model and HGA-SA method, they are applied on a square network, which represents a container port with an annual throughput of 5 million TEUs. A terminal, an empty container depot and two truck depots are contained. Figure 4 is a schematic diagram. It can be seen from Figure 4 that the selected square network is large enough, and the time for trucks to travel along the edge of the network is 160 min. In addition, every experiment will provide the size of the network and the location of the container port. For different transportation companies, the locations of truck depts and the empty container dept are all randomly selected. Customer locations are also randomly determined in the network. The customer's pick-up and delivery time-window is from 4:00 a.m. to 10:00 p.m., while the terminal is open from 8:00 a.m. to 6:00 p.m., which includes 10 time-windows with a duration of one hour. All experiments were performed on a computer equipped with an Intel Core i7, 1.8 GHz CPU and 8 GB RAM.

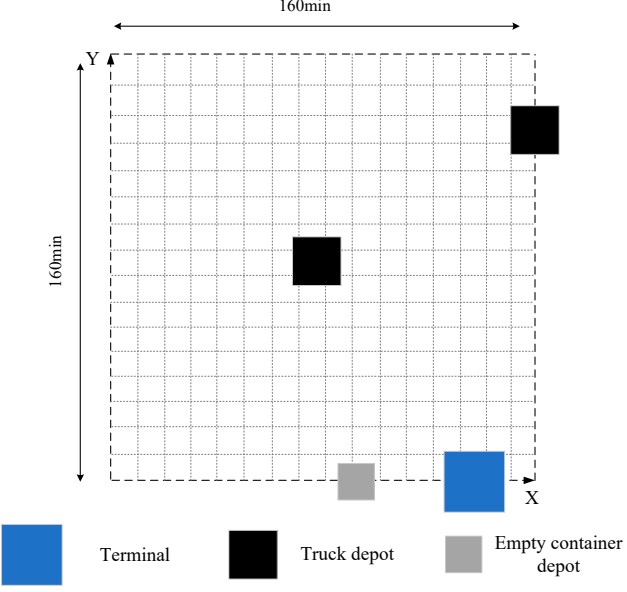

**Figure 4.** Instance network of drayage area.

The basic data used in this work is as follows: the turn time of the truck at the terminal is 43.2 min; the daily working time of the terminal is 10 h, and the number of time-windows is 10 [25]; the terminal average queuing time is 10 min [20,25]; and the time to load/unload the container is 5 min [29].

Each of the five penalty values ranges from 1 to 10 integers, and they are arranged and combined to calculate the objective function value of the unconstrained set of a multi-constraint TAS model under each set of penalty values. The difference between the objective

function values of the constrained set of a multi-constraint TAS model, and the penalty value corresponding to the minimum difference is the optimal penalty value. Finally, the optimal penalty values are determined as $W_l = 1$, $W_s = 3$, $W_p = 1$, $W_n = 3$, and $W_q = 1$.

### 4.1. Computational Experiments and Results

Table 1 shows the experimental results of the designated truck company and the best solution supplied by the proposed TAS. The internal working principle of the multi-constraint TAS model is explained by a few simple experiments. $X_1 \sim X_6$ in Table 1 indicate the unit cost of the factual time-gap is greater than the appointment, the actual time-gap is smaller than the appointment, the actual arrival time-window is later than the appointment, the actual arrival time-window is earlier than the appointment, the average waiting time at the gate, and the congestion time of the morning and evening peak periods, respectively. Experiment 1 involves one truck company making an appointment for a truck. The required arrival time to the terminal is within the time-windows 1, 3, 6, and 8. The best appointment time is during the following time-windows: 1, 4, 6, and 8. The main reason is that the allocation of the time-window 3 is 0, and the appointment of the time-window 3 is extended to the time-window 4. Experiment 2 and Experiment 3 are the scheduling of two trucks. Experiment 3 provides the required time-window. The total target value is the sum of the unit time cost of trucks waiting at the gate and the unit time cost of truck congestion in the morning and evening rush hours.

**Table 1.** The time-window of a truck company and the optimal solution provided by the truck appointment system (TAS).

| Exp. Num. | 1 | | 2 | | 3 | |
|---|---|---|---|---|---|---|
| Quota per time-window | （11,0,1,1,1,1,1,0,1） | | （1,2,1,1,1,2,1,1,1） | | （2,2,2,2,2,2,2,2,2） | |
| Number of jobs | 3 | | 4 | | 5 | |
| Desired arrival time-windows | [1,3,6,8] | | [3,3] | | [1,1] | |
| | | | [2,6,7] | | [3,9,9] | |
| TAS solution | [1,4,6,8] | | [2,3] | | [1,1] | |
| | | | [2,6,7] | | [2,9,9] | |
| Objective value | TOTAL = 12.1678 | $X_1 = 1$ | TOTAL = 17.6862 | $X_1 = 1$ | TOTAL = 23.5191 | $X_1 = 1$ |
| | | $X_2 = 3$ | | $X_2 = 0$ | | $X_2 = 0$ |
| | | $X_3 = 1$ | | $X_3 = 0$ | | $X_3 = 1$ |
| | | $X_4 = 0$ | | $X_4 = 3$ | | $X_4 = 0$ |
| | | $X_5 = 1.9849$ | | $X_5 = 3.2942$ | | $X_5 = 5.2819$ |
| | | $X_6 = 5.1829$ | | $X_6 = 10.3920$ | | $X_6 = 16.2372$ |

Table 2 shows the parameter settings of 28 experiments. According to the experimental parameters in Table 3, the TAS for considering only the gate congestion problem [21], the TAS considering the appointment change cost and the gate congestion problem [22], and the proposed TAS of this study are respectively simulated. Table 3 shows the experimental results. Experiments 4–18 are small-scale problems, experiments 19–25 are medium-scale problems, and experiments 26–31 are large-scale problems (the average amount of containers handled by each truck company exceeds 10 jobs).

**Table 2.** Experimental parameters.

| Exp. Num. | Problem Size | Size (jobs) | Quota Per Time-Window | Num. of Truck Companies | Terminal Coordinates (min) | | Drayage Area(min × min) |
|---|---|---|---|---|---|---|---|
| | | | | | X | Y | |
| 4 | | 4 | (1,1,1,1,1,1,1,1,1,1) | 1 | 0 | 80 | 160 × 160 |
| 5 | | 8 | (2,2,2,2,2,2,2,2,2,2) | 1 | 0 | 80 | 160 × 160 |
| 6 | | 11 | (2,2,3,3,3,3,3,3,2,2) | 2 | 0 | 80 | 160 × 160 |
| 7 | | 15 | (3,3,4,4,4,4,4,4,3,3) | 2 | 0 | 80 | 160 × 160 |
| 8 | | 18 | (4,4,4,4,4,4,4,4,4,4) | 3 | 0 | 80 | 160 × 160 |
| 9 | | 22 | (4,4,5,5,5,5,5,5,4,4) | 3 | 0 | 80 | 160 × 160 |
| 10 | The small scale | 26 | (5,5,6,6,6,6,6,6,5,5) | 4 | 0 | 80 | 160 × 160 |
| 11 | | 32 | (6,6,8,8,8,8,8,8,6,6) | 4 | 0 | 80 | 160 × 160 |
| 12 | | 36 | (7,7,8,8,8,8,8,8,7,7) | 5 | 0 | 80 | 160 × 160 |
| 13 | | 37 | (7,7,9,9,9,9,9,9,7,7) | 5 | 0 | 80 | 160 × 160 |
| 14 | | 50 | (9,9,11,11,11,11,11,11,9,9) | 6 | 0 | 80 | 160 × 160 |
| 15 | | 55 | (10,10,13,13,13,13,13,13,10,10) | 7 | 0 | 80 | 160 × 160 |
| 16 | | 100 | (18,18,22,22,22,22,22,22,18,18) | 13 | 0 | 80 | 160 × 160 |
| 17 | | 157 | (29,29,35,35,35,35,35,35,29,29) | 21 | 0 | 80 | 160 × 160 |
| 18 | | 209 | (38,38,46,46,46,46,46,46,38,38) | 27 | 0 | 80 | 160 × 160 |
| 19 | | 257 | (47,47,57,57,57,57,57,57,47,47) | 32 | 0 | 80 | 160 × 160 |
| 20 | | 324 | (59,59,72,72,72,72,72,72,59,59) | 38 | 0 | 80 | 160 × 160 |
| 21 | | 417 | (76,76,92,92,92,92,92,92,76,76) | 47 | 0 | 80 | 160 × 160 |
| 22 | The medium scale | 492 | (89,89,109,109,109,109,109,109,89,89) | 60 | 0 | 80 | 160 × 160 |
| 23 | | 569 | (103,103,126,126,126,126,126,126,103,103) | 67 | 0 | 80 | 160 × 160 |
| 24 | | 602 | (109,109,133,133,133,133,133,133,109,109) | 71 | 0 | 80 | 160 × 160 |
| 25 | | 663 | (120,120,146,146,146,146,146,146,120,120) | 74 | 0 | 80 | 160 × 160 |
| 26 | | 701 | (131,131,160,160,160,160,160,160,131,131) | 62 | 0 | 80 | 160 × 160 |
| 27 | | 754 | (136,136,166,166,166,166,166,166,136,136) | 70 | 0 | 80 | 160 × 160 |
| 28 | The large scale | 827 | (149,149,182,182,182,182,182,182,149,149) | 75 | 0 | 80 | 160 × 160 |
| 29 | | 1032 | (186,186,228,228,228,228,228,228,186,186) | 82 | 0 | 80 | 160 × 160 |
| 30 | | 1782 | (321,321,393,393,393,393,393,393,321,321) | 90 | 0 | 80 | 160 × 160 |
| 31 | | 2438 | (439,439,537,537,537,537,537,537,439,439) | 157 | 0 | 80 | 160 × 160 |

The small and medium scale problems are solved by lingo software, and the large-scale problem is solved by a hybrid genetic algorithm and simulated annealing. The specific experimental results are shown in Table 3. The truck company hopes to reduce the number of trucks as much as possible to reduce the cost. Through the comparison of the three TASs experiments, it can be seen that the proposed TAS has the least requirements on the number of trucks, which can meet the needs of the truck companies. For the restricted drayage problem, the TAS target value of this study is the smallest, which means that the drayage time is the least. This shows that the TAS makes full use of the time of each truck, which improves the efficiency of the whole port supply chain. Thus, the overall operation cost can be reduced as much as possible to better serve the truck and terminal companies. Considering the impact of the city's morning and evening peak hours on the truck traveling time, a reasonable allocation of the appointment time-windows can reduce the overall operation cost.

*4.2. Comparison and Analysis of Algorithm Performance*

By selecting the benchmark instances of four large-scale truck appointment system schedules as an example, the performances of the HGA-SA and genetic algorithm [25] and reactive Tabu search algorithm [22] are compared. The four examples, which are N1, N2, N3, and N4, are from [22]). The specific performance comparison results are shown in Table 4. It is found that the proposed HGA-SA is superior to the other two algorithms in solving large-scale problems. At the same time, the final target value of the optimal solution obtained by the proposed algorithm is obviously better than the target value obtained by the other two algorithms, which is more in line with the goal of reducing the overall operation cost of the truck companies and the container terminals.

**Table 3.** Experimental result.

| Exp. Num. | TAS Considering only Gate Queuing Time | | | TAS Considering the Change Cost and Gate Waiting Time | | | | | | | | TAS in this Paper | | | | | | | | |
|---|---|---|---|---|---|---|---|---|---|---|---|---|---|---|---|---|---|---|---|---|
| | TAS | Restricted Drayage Problem | | TAS | | | | | | Restricted Drayage Problem | | TAS | | | | | | | Restricted Drayage Problem | |
| | Objective Function Value (Total Cost) | Target Value (min) | Num. of Trucks Required | Objective Function Value (Total Cost) | $X_1$ | $X_2$ | $X_3$ | $X_4$ | $X_5$ | Target Value (min) | Num. of Trucks Required | Objective Function Value (Total Cost) | $X_1$ | $X_2$ | $X_3$ | $X_4$ | $X_5$ | $X_6$ | Target Value (min) | Num. of Trucks Required |
| 4 | 1.3 | 794 | 2 | 3.3 | 1 | 0 | 1 | 0 | 1.3 | 652 | 1 | 3.3 | 1 | 0 | 1 | 0 | 1.3 | 0 | 599 | 1 |
| 5 | 3.5 | 2317 | 3 | 12.3 | 0 | 3 | 0 | 6 | 3.3 | 2143 | 3 | 9.4 | 1 | 0 | 1 | 3 | 3.3 | 1.1 | 2009 | 3 |
| 6 | 4.2 | 3257 | 6 | 23.5 | 1 | 0 | 0 | 18 | 4.5 | 3048 | 4 | 36.2 | 1 | 0 | 9 | 18 | 4.3 | 3.9 | 2731 | 3 |
| 7 | 6.9 | 3249 | 7 | 8.1 | 1 | 0 | 0 | 0 | 7.1 | 2976 | 5 | 27.7 | 1 | 3 | 0 | 9 | 5.3 | 9.4 | 2799 | 4 |
| 8 | 7.9 | 4729 | 9 | 38 | 0 | 6 | 0 | 24 | 8 | 4278 | 7 | 48.6 | 2 | 3 | 3 | 24 | 7.2 | 9.4 | 4258 | 5 |
| 9 | 10.1 | 6031 | 12 | 19.3 | 0 | 0 | 9 | 0 | 10.3 | 5327 | 8 | 32.9 | 1 | 3 | 9 | 0 | 10.5 | 9.4 | 5032 | 8 |
| 10 | 11.2 | 6924 | 12 | 35.6 | 0 | 0 | 0 | 24 | 11.6 | 6643 | 9 | 54.5 | 0 | 6 | 0 | 18 | 10.9 | 19.6 | 6337 | 8 |
| 11 | 16.8 | 8541 | 13 | 43.1 | 0 | 9 | 0 | 18 | 16.1 | 8132 | 13 | 73.5 | 1 | 9 | 3 | 24 | 16.9 | 19.6 | 7149 | 11 |
| 12 | 16.3 | 9189 | 15 | 49.4 | 2 | 6 | 4 | 21 | 16.4 | 8574 | 14 | 70 | 3 | 3 | 9 | 18 | 17.4 | 19.6 | 8142 | 12 |
| 13 | 17.9 | 1,0145 | 18 | 86.2 | 2 | 15 | 0 | 51 | 18.2 | 9721 | 14 | 89.4 | 3 | 15 | 3 | 27 | 19.3 | 22.1 | 9130 | 13 |
| 14 | 22.8 | 1,5011 | 22 | 62.9 | 1 | 9 | 0 | 30 | 22.9 | 1,3829 | 20 | 104.7 | 0 | 9 | 3 | 33 | 22.8 | 36.9 | 1,3580 | 18 |
| 16 | 47.3 | 2,7398 | 45 | 101.7 | 0 | 12 | 1 | 45 | 43.7 | 2,6493 | 33 | 147.4 | 1 | 6 | 3 | 51 | 46.9 | 39.5 | 2,2671 | 27 |
| 17 | 86.7 | 5,1147 | 96 | 180.2 | 0 | 21 | 0 | 72 | 87.2 | 5,0436 | 60 | 239.4 | 0 | 24 | 3 | 57 | 82.6 | 72.8 | 4,3792 | 42 |
| 18 | 111.2 | 6,1985 | 107 | 235.3 | 10 | 36 | 0 | 78 | 111.3 | 6,0654 | 80 | 292.2 | 7 | 24 | 1 | 75 | 105.6 | 79.6 | 5,8430 | 69 |
| 19 | 127.3 | 6,9275 | 123 | 286 | 14 | 42 | 0 | 92 | 138 | 6,7528 | 89 | 386.9 | 12 | 45 | 1 | 87 | 131.7 | 110.2 | 6,6801 | 83 |
| 20 | 151.7 | 8,3892 | 136 | 350.9 | 19 | 54 | 0 | 126 | 151.9 | 8,1467 | 109 | 494 | 14 | 51 | 3 | 153 | 153.3 | 119.7 | 7,5350 | 89 |
| 21 | 213.7 | 11,7644 | 159 | 400 | 1 | 48 | 0 | 164 | 187 | 11,7257 | 136 | 724.9 | 12 | 39 | 1 | 195 | 219.4 | 258.5 | 11,4942 | 132 |
| 22 | 231.9 | 13,3572 | 173 | 457.3 | 0 | 36 | 0 | 189 | 232.3 | 13,2672 | 161 | 834.4 | 0 | 33 | 1 | 243 | 226.2 | 331.2 | 11,9985 | 152 |
| 23 | 496.1 | 17,6785 | 301 | 702.2 | 0 | 42 | 1 | 182 | 477.2 | 15,9638 | 213 | 853.4 | 0 | 27 | 0 | 270 | 229.4 | 327 | 12,9041 | 198 |
| 24 | 602.7 | 18,9064 | 324 | 963.6 | 0 | 75 | 0 | 275 | 613.6 | 16,7382 | 227 | 1309.5 | 0 | 45 | 0 | 309 | 459.3 | 496.2 | 15,7029 | 210 |
| 25 | 662.4 | 19,3926 | 336 | 1028.1 | 0 | 75 | 0 | 291 | 662.1 | 17,3083 | 235 | 1640.3 | 0 | 51 | 1 | 393 | 594.1 | 601.2 | 17,8592 | 232 |
| 26 | 764.9 | 22,6839 | 342 | 1190.8 | 0 | 68 | 1 | 375 | 746.8 | 20,1126 | 254 | 1782.6 | 0 | 54 | 3 | 471 | 632.1 | 622.5 | 18,2952 | 227 |
| 27 | 874.7 | 26,7538 | 367 | 1319.9 | 1 | 89 | 0 | 423 | 806.9 | 22,6875 | 287 | 1878.2 | 12 | 54 | 0 | 492 | 647.8 | 672.4 | 23,0601 | 249 |
| 28 | 942.8 | 29,6423 | 376 | 1404.9 | 0 | 72 | 1 | 468 | 863.9 | 24,7532 | 307 | 1900.1 | 0 | 61 | 0 | 537 | 622.8 | 679.3 | 23,9024 | 276 |
| 29 | 1267.9 | 40,7743 | 487 | 1814.8 | 6 | 87 | 0 | 587 | 1134.8 | 39,8631 | 427 | 2247.4 | 0 | 75 | 0 | 590 | 698.3 | 884.1 | 34,9529 | 396 |
| 30 | 1246.9 | 43,8649 | 507 | 2174.7 | 8 | 92 | 0 | 621 | 1453.7 | 41,5478 | 447 | 2668.2 | 0 | 93 | 3 | 652 | 938.1 | 982.1 | 38,4750 | 419 |
| 31 | 2658.8 | 93,5753 | 924 | 4230.9 | 0 | 187 | 0 | 1067 | 2976.9 | 85,7228 | 874 | 4744.6 | 8 | 162 | 1 | 1019 | 1731.6 | 1823 | 82,9420 | 776 |

**Table 4.** Algorithm performance comparison.

| | | | HGA-SA | | GA | | RTSA | |
|---|---|---|---|---|---|---|---|---|
| Exp. Num. | Number of Jobs | Num. of Truck Companies | Target Value (min) | CPU(s) | Target Value (min) | CPU(s) | Target Value (min) | CPU（s) |
| N1 | 582 | 50 | 17,2948 | 7249.92 | 23,4596 | 8032.47 | 20,3495 | 7639.29 |
| N2 | 582 | 40 | 20,2950 | 7439.47 | 24,3858 | 7932.59 | 20,4950 | 8639.59 |
| N3 | 582 | 10 | 25,4893 | 1,0060.27 | 32,4560 | 1,0729.48 | 30,3028 | 9932.83 |
| N4 | 582 | 5 | 26,3458 | 9438.28 | 35,3940 | 1,0183.38 | 33,4567 | 1,1372.62 |

To verify that the performance of the proposed HGA-SA, based on the parameters of Experiment 27, the above three algorithms are run continuously 10 times. Comparisons of box plots for convergence time, convergence algebra, and fitness are shown in Figure 5. Algorithms 1–33 are HGA-SA, Genetic algorithm (GA, Shiri and Huynh [25]), and Reactive Tabu Search algorithm (RTSA, Mohammad et al. [26]), respectively.

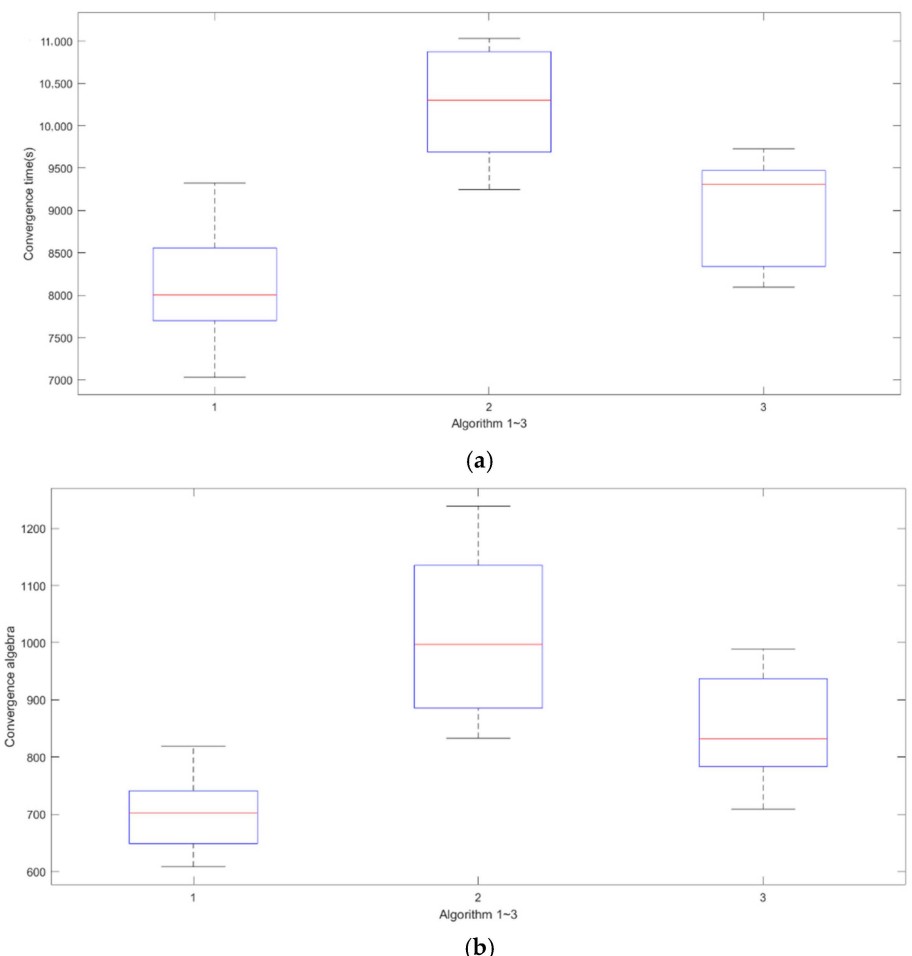

**Figure 5.** *Cont.*

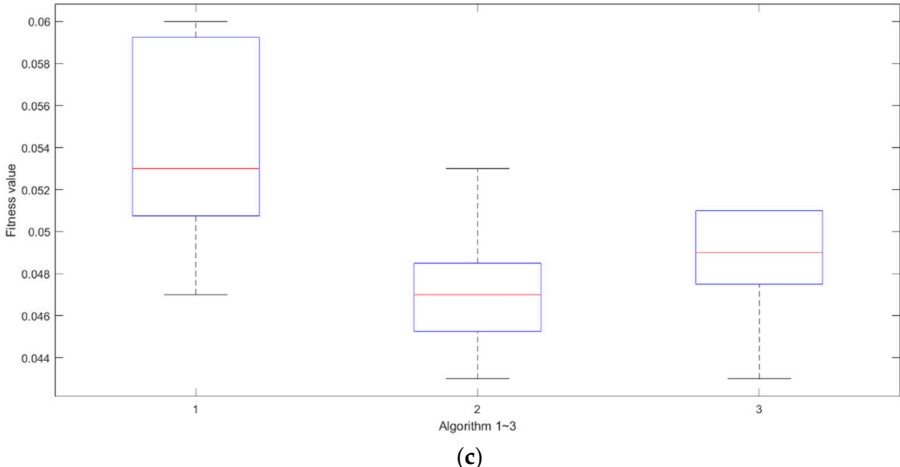

(**c**)

**Figure 5.** Comparison of algorithm optimization performances ((**a**): Convergence time comparison; (**b**): Convergence algebra comparison; (**c**): Fitness value comparison).

Figure 5 shows that the optimization results and optimization speed of the proposed method are better than the other two algorithms. This is because the HGA-SA algorithm aims at the proposed multi-constraint TAS model and makes full use of the Equation (13) threshold concept to set a special iterative method, which can reduce unnecessary steps. At the same time, by combining the advantages of the simulated annealing algorithm and genetic algorithm, the search intensity is increased and the accuracy of the solution is enhanced.

### 4.3. Comparison and Analysis of Operation Cost

4.3.1. Impact of Customer Time-Windows on Operation Cost

In order to test the influence of customer time-windows on the overall operation cost, five experiments with different time-windows are performed, and the multi-constraint TAS in this work is compared with the TASs of Chen et al. [21] and Mohammad et al. [22]. Figure 6 shows the comparative experimental results of the three TASs. As shown in Figure 6, the customer time-window is longer and the operation cost is lower. If the customer time-window is shortened, the port workload will increase with the requirement of more devices and better productivity. The multi-constraint TAS in this work considered the impact of the morning and evening rush hours on the arrival time of trucks, and allocated the reserved trucks more reasonably; the overall operation cost was lower than that of the other two TASs. For the port company, reasonable planning of the customer time-window can effectively reduce the overall operation cost.

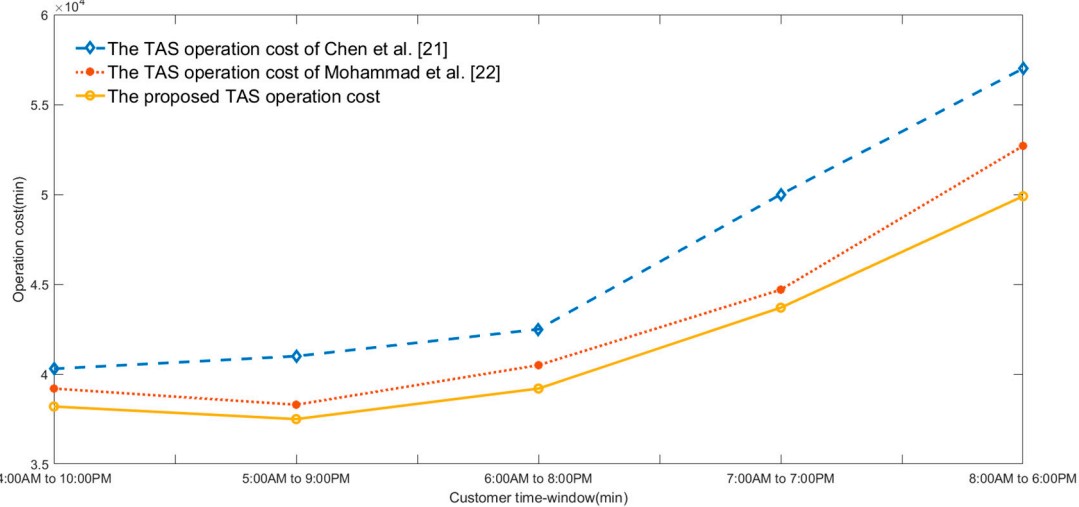

**Figure 6.** TAS Operation costs under Different Customer Time-Window.

### 4.3.2. Impact of Terminal Time-Window Duration on Operation Cost

The number of the terminal time-windows is 10, and the time-window duration is 1 h. To analyze the influence of different terminal time-window durations on the operation costs, three experiments are implemented. Figure 7 shows the results. For three TASs, extending the terminal time-window duration results in lower operation costs. For the terminal companies, the shorter the duration of each time-window, the smaller the order quantity assigned by the freight forwarder and the smaller the truck appointment quantity will be. For the truck companies, the long time-window duration reduces operation costs. However, considering the uncertainties, such as changes in the transportation path, longer duration is more advantageous for the TASs. Compared with the TAS operation cost of Chen et al. [21], the operation costs of the multi-constraint TAS and Mohammad et al. [22] are small. This is because the duration of the time-window is large enough, the operation cost considered by the TAS of Mohammad et al. [22] is for the entire time-window. And the multi-constraint TAS in this work considers the impact on the truck travel time in morning and evening peak congestion and is aimed at the time-window close to the morning and evening peak hours. When the time-window duration is greater than 120 min, the operation costs of Mohammad et al. [22] and the multi-constraint TAS are hardly changed. Owing to the adequate terminal time-window duration, the impact of morning and evening peak on the arrival time of container trucks will be smaller. Moreover, the possibility of trucks not arriving at the port on time will be reduced, and the cost of changing the appointment plan will also be reduced, which benefits both the truck companies and the terminal companies. For different terminal time-windows duration, the overall operation cost of multi-constraint TAS is the lowest.

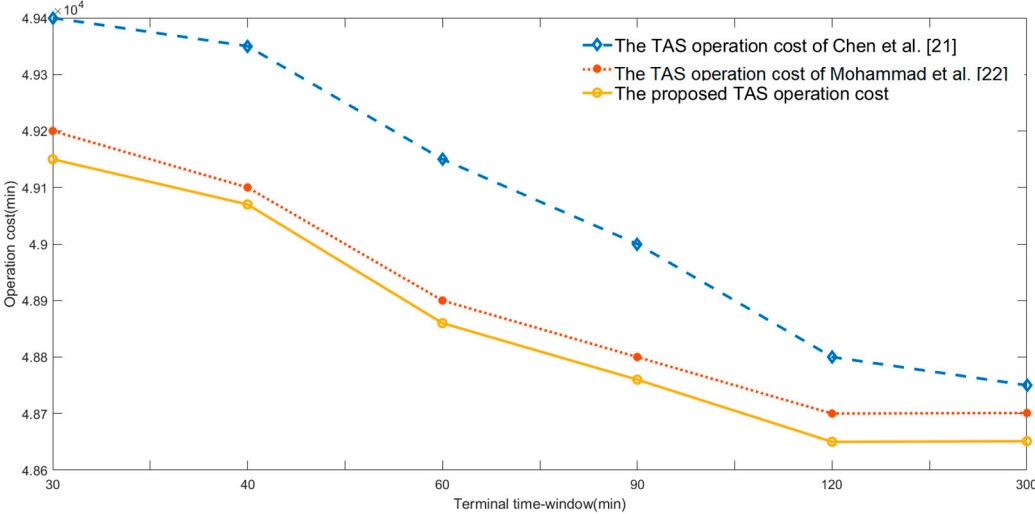

**Figure 7.** TAS Operation Cost under Different Terminal Time-Window Duration.

### 4.3.3. Impact of Terminal Turn Time on Operation Cost

To analyze the effect of the terminal turn time on the overall operation cost, 5 different experiments are implemented and compared by dividing the terminal turn time into five groups (20 min, 30 min, 40 min, 50 min, 60 min). The result is shown in Figure 8. The operation cost increases with the increase of the terminal turn time. However, when the terminal turn time is close to 60 min, the operation costs of each TAS are almost the same. Therefore, when the terminal turn time continues to increase, the impact of the terminal turn time on the operation cost decreases. For each TAS, the longer the terminal turn time, the higher the operation cost, the lower the operation efficiency of the port. For TASs, in different studies, the multi-constraint TAS also considers the congestion in the morning and evening rush hours. Under the condition of same terminal turn time, the multi-constraint

TAS has the lowest operation cost. Therefore, the morning and evening peaks have a non-negligible impact on the decision-making of TAS. For the terminal company, reasonable setting turn time can effectively reduce operation cost and serve the customer better.

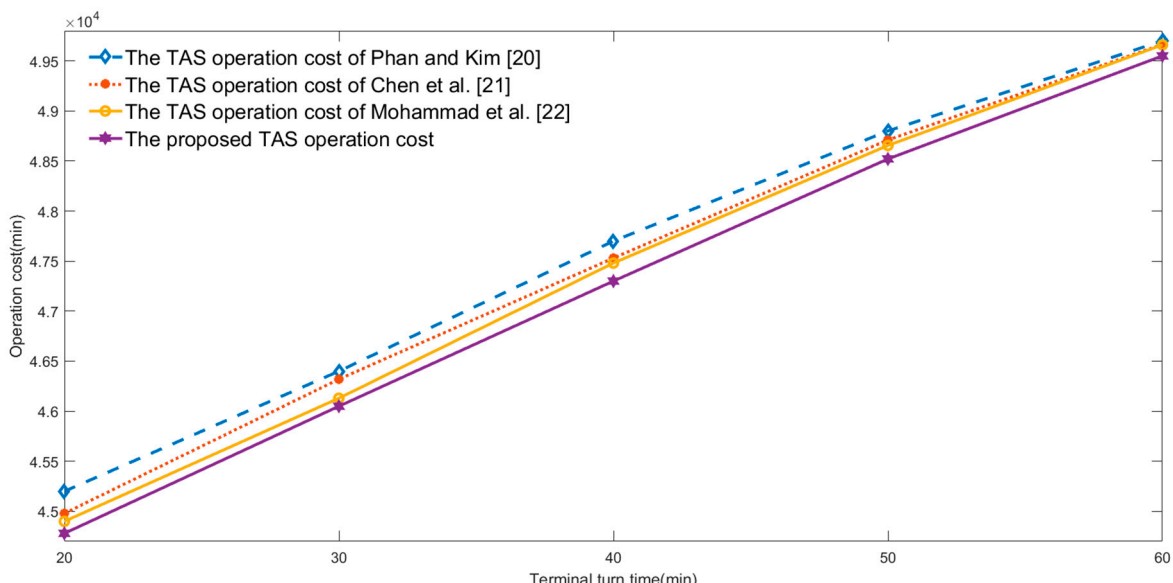

**Figure 8.** TAS Operation Cost under Different Terminal Turn Time.

## 5. Conclusions

Aiming at eliminating the port congestion at gates and yards, this research proposes a multi-constraint TAS model and a hybrid genetic algorithm and simulated annealing (HGA-SA) method considering morning and evening peak congestion. The most remarkable feature of our multi-constraint TAS model is that we use mixed integer nonlinear programming to determine the best appointment plan for each truck with the target of minimizing the overall operation cost. A new iterative method is proposed, and the mutation probability is autonomously adjusted according to the evolutionary algebra, which significantly reduces the computation time of the HGA-SA. The results obtained on benchmark instances from the literature show the good performance of our HGA-SA method. Our multi-constraint TAS model can even provide lower operation costs than the TASs of Chen et al. [21] and Mohammad et al. [22] for different customer time-windows, terminal time-window duration, and terminal turn time instances. The tests on benchmark instances show that our multi-constraint TAS model and HGA-SA method provide better solutions, which benefit both the truck companies and the terminal companies.

From a management perspective, our research is of great significance. The use of the proposed multi-constraint TAS allows the terminal company to formulate reasonable treatments for each truck company based on its amount and completion, which results in a higher satisfaction for the customers. At the same time, the terminal company can allocate the quantity of containers more reasonably and have a stronger grasp of the arrival time of the truck. The most important thing is that the implementation of a multi-constraint TAS can make the truck reach the terminal more evenly, thus effectively controlling congestion in consideration of the sustainable development of ports. The model and method developed in the work can be used by different stakeholders of port supply chains, such as projecting organizations and port authorities, to justify the decisions for increasing the performance of container terminals, lowering the operation cost, and servicing customers better.

As a future study, a natural extension of this work would develop acceleration techniques, such as sophisticated heuristic algorithms, for obtaining higher quality solutions than the HGA-SA method in a shorter computation time. Another interesting extension of our current work would be to enhance the proposed multi-constraint TAS model with the

consideration of uncertainties (such as severe weather, traffic accident, late or early arrival of truck), multi-objectives (such as operation time, facility utilization, freight demand satisfaction rate, energy consumption) and so on. We believe that our multi-constraint TAS model can be a used as a general framework since it is feasible to incorporate other constraints and objectives into it.

**Author Contributions:** Conceptualization, B.X. and X.L.; methodology, B.X., J.L., and X.L.; validation, B.X., J.L., and X.L.; writing—review and editing, B.X., J.L., and X.L.; supervision, Y.Y. and O.P. All authors have read and agreed to the published version of the manuscript.

**Funding:** This work was supported by National Social Science Foundation project of China (No. 18BGL109).

**Institutional Review Board Statement:** Not applicable.

**Informed Consent Statement:** Not applicable.

**Data Availability Statement:** Data is contained within the article.

**Conflicts of Interest:** The authors declare that they have no conflicts of interest regarding the publication of this article.

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
