# Peer review of "Optimization for a Multi-Constraint Truck Appointment System Considering Morning and Evening Peak Congestion"

_sustainability, doi:10.3390/su13031181_

Round 1
Reviewer 1 Report
Review 15.12. JRFM
I want to thank the authors for submitting their very interesting paper for this journal. The value of the research is high, but still, the manuscript should be amended.
Please follow my comments (in chronological order of your paper).
- Content of the manuscript
- The abstract is acceptable, but I would advise adding one sentence about the impact and/or managerial implications
- Introduction: There is a need to include a clear statement of the aim of the study (obligatory) and research questions (not obligatory).
- Line 98 – the Figure is not clear without the detailed description. Please add the description for this figure.
- Section 2.1. – this is only a Table without caption and description. Please amend it.
- Additionally, please check if all tables and figures have captions in the text and clear description.
- Method and numerical experiments are presented clearly. The only thing is to proofread them.
- Summary is too short. There should be at least some initial discussion – there is no discussion at all, and the problem is crucial for the sector. The part of the summary is too similar to the abstract and other parts of the paper (paragraph 2). Limitations are described in a very limited way, and this is not enough. How the authors addressed or solved those limitations? Are there any future research plans or directions? What is the impact – filling some gaps mentioned in the introduction?
- Technical layer:
- Line 14 – based on?
- Please revise the text to avoid small typos and similar (e.g. line 34, 36, 178)
use the template of the journal to prepare your manuscript in the right format: https://www.mdpi.com/journal/jrfm/instructions
- Some parts of the paper are missing like Author Contributions, Funding
- Citations in text and references are made in the wrong format.
- Line 39 -the source for the data is needed
- Line 43-44 – mid 21st century is not proper here
- the whole paper should be rewritten or proofread because it is hard to understand some parts of the text. The flow is weak; therefore, sometimes following the research path in research framework is difficult.
- Line 121 – should be „seek”
- Line 354 - ?
- 367-368 – not clear sentence
- Should the names of the authors be written starting with capital letters?
To sum up, the paper is well-structured, well-prepared but needs major revisions. I am looking forward to seeing the improved text.
Author Response
Reply to Reviewer#1
I want to thank the authors for submitting their very interesting paper for this journal. The value of the research is high, but still, the manuscript should be amended. Please follow my comments (in chronological order of your paper).
- Content of the manuscript
Response: Thanks for your comments. We have completely revised the content of the article to make the whole article look more logical. All modified parts are shown in red.
- The abstract is acceptable, but I would advise adding one sentence about the impact and/or managerial implications
Response: Thanks for your comments. At the end of the abstract, we have summarized the comparison between the proposed TAS and other conventional TASs, and pointed out that the proposed TAS can not only better serve truck companies and container terminals but also more effectively reduce their overall operation cost compared with the traditional TASs. (The details are in the lines 22 to 24 of the manuscript).
- Introduction: There is a need to include a clear statement of the aim of the study (obligatory) and research questions (not obligatory).
Response: Thanks for your comments. We have rewritten the introduction section. Not only clearly pointed out the purpose of the research (The details are in lines 94 to 102 in the manuscript), but also increased the relevant literature to highlight the necessity of our research (The details are in lines 46 to 93 in the manuscript).
- Line 98 – the Figure is not clear without the detailed description. Please add the description for this figure.
Response: Thanks for your comments. We have added the description for the Figure 1 (The details are in lines 120 to 125 in the manuscript). And we have carefully checked the article to ensure that all tables and figures are described in detail.
- Section 2.1. – this is only a Table without caption and description. Please amend it.
Response: Thanks for your comments. We are sorry for the misunderstanding of this part due to the presentation problem. Section 2.1 is indices, parameters and sets for a multi-constrained TAS. We have added the description of this part and adjusted the part of the symbol expression.
- Additionally, please check if all tables and figures have captions in the text and clear description.
Response: Thanks for your comments. We have checked all tables and figures to ensure that they have captions in the text and clear description.
- Method and numerical experiments are presented clearly. The only thing is to proofread them.
Response: Thanks for your comments. We have proofread the method and numerical experiments.
- Summary is too short. There should be at least some initial discussion – there is no discussion at all, and the problem is crucial for the sector. The part of the summary is too similar to the abstract and other parts of the paper (paragraph 2). Limitations are described in a very limited way, and this is not enough. How the authors addressed or solved those limitations? Are there any future research plans or directions? What is the impact – filling some gaps mentioned in the introduction?
Response: Thanks for your comments. We have reorganized the summary section. The content of the summary is enriched and the management significance of this research is added (The details are in lines 401 to 423 in the manuscript). The content of the original paragraph 2 has also been revised to describe the purpose of this study more clearly (The details are in lines 94 to 102 in the manuscript). And in the last paragraph of the summary, the limitations of this article and future work are added (The details are in lines 424 to 431 in the manuscript).
- Technical layer.
Response: Thanks for your comments. On the technical layer, we have carefully checked and modified related models and methods based on expert opinions.
- Line 14 – based on?
Response: Thanks for your comments. We have revised the content of line 14 and checked the full text to ensure that there are no writing and grammatical errors.
- Please revise the text to avoid small typos and similar (e.g., line 34, 36, 178)
Response: Thanks for your comments. We have revised the text to avoid small typos and similar.
- Use the template of the journal to prepare your manuscript in the right format: https://www.mdpi.com/journal/jrfm/instructions.
Response: Thanks for your comments. We have revised our manuscript in the right format.
- Some parts of the paper are missing like Author Contributions, Funding.
Response: Thanks for your comments. We have added Author Contributions and Funding, and have adjusted the structure of the full text strictly in accordance with the requirements of the journal (The details are in lines 432 to 435 in the manuscript).
- Citations in text and references are made in the wrong format.
Response: Thanks for your comments. We have revised citations in text and references in the right format and ensured that the full text format is uniform.
- Line 39 -the source for the data is needed.
Response: Thanks for your comments. We have added the source for the data of line 39(The details are in line 42 in the manuscript).
- Line 43-44 – mid 21st century is not proper here
Response: Thanks for your comments. We have revised the inappropriate expression (The details are in lines 46 to 48 in the manuscript).
- the whole paper should be rewritten or proofread because it is hard to understand some parts of the text. The flow is weak; therefore, sometimes following the research path in research framework is difficult.
Response: Thanks for your comments. We have revised the entire structure of the article. The whole article is more logical and more contextual. In addition, a careful proof reading has been done, and “Editage” has helped us to make the English expression better. The following is certificate of editing. In this way, we have tried our best to make it more readable.
- Line 121 – should be „seek”
Response: Thanks for your comments. We have revised the content of line 121 and checked the full text to ensure that there are no writing and grammatical errors (The details are in lines 148 to 149 in the manuscript).
- Line 354 -?
Response: Thanks for your comments. We have revised the content of line 354 (The details are in line 384 in the manuscript).
- 367-368 – not clear sentence
Response: Thanks for your comments. We have revised the sentences from line 367 to line 368 to make the entire sentence clearer (The details are in lines 395 to 397 in the manuscript).
- Should the names of the authors be written starting with capital letters?
Response: Thanks for your comments. We have revised the author’s name writing format in accordance with the journal’s requirements (The details are in line 5 in the manuscript).

Reviewer 2 Report
This paper develops an optimization model for a gate appointment system for a container terminal. The model is tested on several scenarios with various complexity and the results are well presented. The level of English is satisfactory and the organization of the paper is well-structured.
What I would recommend to the authors is to improve the literature review and the theoretical background. The paper falls short on providing enough references to:
- Indicate the research gap
- emphasis on the originality of the research
- Provide the details of how the correct research differs from other similar papers.
I recommend the authors to add more citations in the literature review part. They may add more research about container terminal gate operations and congestions, gate and truck appointment systems, and emphasize how the current research improves upon the cited research.
Author Response
Reply to Reviewer#2
This paper develops an optimization model for a gate appointment system for a container terminal. The model is tested on several scenarios with various complexity and the results are well presented. The level of English is satisfactory and the organization of the paper is well-structured.
What I would recommend to the authors is to improve the literature review and the theoretical background. The paper falls short on providing enough references to:
- Indicate the research gap
Response: Thanks for your comments. We have added more literature and related literature descriptions in the introduction section, which not only highlights the advantages of similar literature (The details are in lines 46 to 93 in the manuscript), but also points out the limitations of these literatures, and makes comparisons with the fundamental research. The penultimate paragraph of the introduction summarizes the innovations of our research, highlighting the purpose and importance of the research (The details are in lines 94 to 102 in the manuscript). At the same time, we summarized the innovation, put forward corresponding management suggestions and future research directions (The details are in lines 401 to 423 in the manuscript).
- emphasis on the originality of the research
Response: Thanks for your comments. At the end of the abstract, we have pointed out that the proposed TAS is more suitable for truck companies and container terminals (The details are in lines 22 to 24 in the manuscript). Then we added a comparison between related literature and this article in the introduction, highlighting the innovation of this article (The details are in lines 46 to 93 in the manuscript). Finally, we added a preliminary discussion of the results and the management significance derived from the sensitivity analysis in the comparison and analysis of operation cost section, highlighting the advantages of this research (The details are in lines 401 to 423 in the manuscript).
- Provide the details of how the correct research differs from other similar papers.
Response: Thanks for your comments. We have added the corresponding comparative results and management significance to the abstract to better highlight the superiority of the study (The details are in lines 22 to 24 in the manuscript). Then we added a comparison between related literature and this article in the introduction, highlighting the innovation of this article (The details are in lines 46 to 93 in the manuscript). Finally, we added a preliminary discussion of the results and the management significance derived from the sensitivity analysis in the comparison and analysis of operation cost section, highlighting the advantages of this research (The details are in lines 401 to 423 in the manuscript).
Reviewer 3 Report
Thank You for the invitation to review this paper.
The paper is quite innovative and well written using academic standards.
Please consider to extend literature review. Maybe some sources will be interesting in this area:
Kot, S. Cost structure in relation to the size of road transport enterprises [Struktura kosztów w relacji do wielkoĹ›ci przedsiÄ™biorstw transportu drogowego] (2015) Promet - Traffic - Traffico, 27 (5), pp. 387-394. Pandian, S.R., Soltysova, Z.Management of mass customized orders using flexible schedules to minimize delivery times. (2018) Polish Journal of Management Studies, 18 (1), pp. 252-261.
Author Response
Reply to Reviewer#3
Thank You for the invitation to review this paper.
The paper is quite innovative and well written using academic standards.
Please consider to extend literature review. Maybe some sources will be interesting in this area:
Kot, S. Cost structure in relation to the size of road transport enterprises [Struktura kosztów w relacji do wielkoĹ›ci przedsiÄ™biorstw transportu drogowego] (2015) Promet - Traffic - Traffico, 27 (5), pp. 387-394.
Pandian, S.R., Soltysova, Z.Management of mass customized orders using flexible schedules to minimize delivery times. (2018) Polish Journal of Management Studies, 18 (1), pp. 252-261.
Response: Thanks for your comments. We carefully read the literature recommended by you, and made an explanation in the study (Details are in lines 61 to 62 and 90 to 93 of the manuscript), which provided new ideas for our research.
We appreciate for Editors/Reviewers’ warm work earnestly, and hope that the correction will meet with approval. Once again, thank you very much for your comments and suggestions.
Round 2
Reviewer 1 Report
I want to thank the authors for improving the paper substantially, according to mine and other reviewers’ comments. Now the manuscript is far better and needs only some minor amendment to meet my requirements. Therefore, I request only one technical issue:
- Still, I want to force the idea of clearly stating the aim as the separate part of the introduction. You may rebuild a little the paragraph between lines 94 and 102 and add the clear indication with one sentence with the aim of the study (not the aim of the model). ("The aim of the study is...").
All the remaining comments from my previous review were addressed, both in the technical and content layer. Well done!
Author Response
Dear Editor and Reviewers:
On behalf of authors, we thank you very much for giving us an opportunity to improve our manuscript; we appreciate editor and reviewer’s effort to provide positive and constructive comments and suggestions on our manuscript (Manuscript ID: sustainability-1051979).
We have studied comments carefully and have made correction which we hope meet your requirements. The main corrections of the revised paper and the answer to the reviewer’s comments are as flowing presented.
Reply to Reviewer#1
I want to thank the authors for improving the paper substantially, according to mine and other reviewers’ comments. Now the manuscript is far better and needs only some minor amendment to meet my requirements. Therefore, I request only one technical issue:
- Still, I want to force the idea of clearly stating the aim as the separate part of the introduction. You may rebuild a little the paragraph between lines 94 and 102 and add the clear indication with one sentence with the aim of the study (not the aim of the model). ("The aim of the study is...").
All the remaining comments from my previous review were addressed, both in the technical and content layer. Well done!
Response: Thanks for your comments. We have rebuilt the original paragraph between lines 94 and 102 and added the clear indication with two sentences with the aim of the study (“the aim of the study is to develop a higher-quality TAS with improved rationality and effectiveness. Specifically, it can better determine the truck’s appointment time-window, lessen the impact of adjustment on the truck company's expected appointment plans, mitigate the queue time of the truck at gates, and meet the order demand of the container terminals.”). In addition, we added a sentence (“In summary, previous studies on TAS mainly considered the queue length of trucks at the gate, and few studies considered the interests of both truck companies and port companies, and the impact of morning and evening peak traffic on the time of truck arrival at the port.”) at the beginning of this paragraph to point out the current research gaps in this field and highlight our study aim.